# Chemistry and Bioactivity of *Microsorum scolopendria* (Polypodiaceae): Antioxidant Effects on an Epithelial Damage Model

**DOI:** 10.3390/molecules27175467

**Published:** 2022-08-25

**Authors:** Cristóbal Balada, Valentina Díaz, Mónica Castro, Macarena Echeverría-Bugueño, María José Marchant, Leda Guzmán

**Affiliations:** 1Laboratorio de Biomedicina y Biocatálisis, Instituto de Química, Facultad de Ciencias, Pontificia Universidad Católica de Valparaíso, Avenida Universidad 330, Valparaíso 2340000, Chile; 2Laboratorio de Propagación, Escuela de Agronomía, Facultad de Ciencias Agronómicas y de los Alimentos, Pontificia Universidad Católica de Valparaíso, La Palma S/N, Quillota 2260000, Chile; 3Laboratorio de Patología de Organismos Acuáticos y Biotecnología Acuícola, Facultad de Ciencias de la Vida, Universidad Andrés Bello, Quillota 980, Viña del Mar 2531015, Chile; 4Centro Interdisciplinario para la Investigación Acuícola (INCAR), Universidad Andrés Bello, Quillota 980, Viña del Mar 2531015, Chile

**Keywords:** antioxidant, anti-inflammatory, Matu’a Pua’a, polyphenols, flavonoids, Rapa Nui

## Abstract

*Microsorum scolopendia* (*MS*), which grows on the Chilean island of Rapa Nui, is a medicinal fern used to treat several diseases. Despite being widely used, this fern has not been deeply investigated. The aim of this study was to perform a characterization of the polyphenolic and flavonoid identity, radical scavenging, antimicrobial, and anti-inflammatory properties of *MS* rhizome and leaf extracts (RAE and HAE). The compound identity was analyzed through the reversed-phase high-performance liquid chromatography (RP-HPLC) method coupled with mass spectrometry. The radical scavenging and anti-inflammatory activities were evaluated for DPPH, ORAC, ROS formation, and COX inhibition activity assay. The antimicrobial properties were evaluated using an infection model on Human Dermal Fibroblast adult (HDFa) cell lines incubated with *Staphylococcus aureus* and *Staphylococcus epidermidis*. The most abundant compounds were phenolic acids between 46% to 57% in rhizome and leaf extracts, respectively; followed by flavonoids such as protocatechic acid 4-O-glucoside, cirsimaritin, and isoxanthohumol, among others. *MS* extract inhibited and disaggregated the biofilm bacterial formed and showed an anti-inflammatory selective property against COX-2 enzyme. RAE generated a 64% reduction of ROS formation in the presence of *S. aureus* and 87.35% less ROS in the presence of *S. epidermidis* on HDFa cells. *MS* has great therapeutic potential and possesses several biological properties that should be evaluated.

## 1. Introduction

*Microsorum scolopendria* (*MS*) is a fern that grows on the Chilean island of Rapa Nui and is known to the indigenous people as “Matu’a Pua’a.” *MS* is native to Polynesian islands, including Fiji, Tahiti, Hawaii, Rapa Nui, and Madagascar [1], where it is used to treat asthma, inflammatory diseases, and cancer [2]. *MS* is known under different names, among them *Phymatosorus scolopendria*, *Polypodium scolopendria*, and *Microsorum parksii* [3,4], *Polypodium scolopendria*, and *Microsorum parksii* [3,4]. As a member of the *Polypodiaceae* family, *MS* has a creeping rhizome 5–7 mm in diameter and leathery-textured leaves with visible veins [5]. Many of the properties attributed to ferns (and plants in general) have been associated with high contents of polyphenols in plant tissues [6,7,8]. More than eight thousand phenolic compounds have been reported in various plant species [9,10]. Polyphenols are widely used in the pharmaceutical, cosmetic, and food industries [11], mainly due to their properties as antioxidants, antimicrobial agents, and inhibitors of pro-inflammatory mediators [12]. There is some diversity in the structure of polyphenols, which has led to their subdivision into families centered on phenolic acids, stilbenes, and flavonoids.

*MS* has not been extensively investigated through phytochemical analyses or characterization of its medicinal properties, despite being widely used by the indigenous people of Rapa Nui. A MEDLINE search using the terms “*Polypodium scolopendria* or *Microsorum scolopendria*” yielded only three articles in 21 years (between 2000 and 2021). Thus, *MS* has been very under-researched so far, only some studies have explored its ecdysteroid content [3]. Due to its importance in traditional Rapa Nui medicine, where it is used to treat a wide variety of ailments, the number of specimens is declining. In fact, the conservation status of *MS* is now classified as “vulnerable” in Chile. It is important to investigate whether it is safe to consume fern material as part of folk medicine, given that certain compounds can be toxic if ingested in large quantities. Thus, the purpose of this study was to perform a functional characterization of Rapa Nui *MS* rhizome and leaf extracts with respect to their polyphenolic and flavonoid identity, radical scavenging, antimicrobial, and anti-inflammatory properties. To this end, the reversed-phase high-performance liquid chromatography (RP-HPLC) method coupled with mass spectrometry was used [13]. The radical scavenging capacity of the *MS* extracts was determined by the DPPH (1,1-diphenyl-2-picrylhydrazyl) radical scavenging method and the oxygen radical absorbance capacity (ORAC) method. The antimicrobial activity against *Staphylococcus aureus* and *Staphylococcus epidermidis* was evaluated in an infection model using the Human Dermal Fibroblast adult (HDFa) cell line. The two bacteria are frequently associated with infections in implants, peripheral venous catheters, and skin [14,15].

Bacterial infections, pathologies caused by UV exposure, and cancer share a common inflammatory process, triggered by the activation and downregulation of several signaling pathways, in which reactive oxygen species (ROS) induce certain cellular disorders. Signal transduction in conditions involving ROS-mediated metabolic activity, inflammatory activity, and metabolic dysfunction is, therefore, an important therapeutic focus for the control of various diseases [16]. In addition, we demonstrated that these extracts have anti-inflammatory, radical scavenging capacity, and sun protection properties. 

## 2. Results

### 2.1. Characterization of the Principal Chemical Components in MS Extracts

#### 2.1.1. Determination of Secondary Metabolites

The results presented in Table 1 show the mass (mg) of the dry extract obtained from each sample evaluated per g of sample. The leaf extracts treated with ethyl acetate were found to have higher yields in mg/g. 

For both types of extracts, polyphenol content was determined using gallic acid (GAE) as an internal standard [17]. Total flavonoid content was determined using kaempferol (KE) as an internal standard, because of the higher sensitivity of flavanols to this technique [18]. The results were expressed as mg GAE/g dry sample for all polyphenols analyzed and mg KE/g dry sample for flavonoids, as shown in Table 2.

The results show that RAE, which corresponds to the rhizome extract, contains more total polyphenols and total flavonoids than HAE.

#### 2.1.2. RP-HPLC-*MS*/*MS* of *MS* Extracts

Reversed-phase high-performance liquid chromatography (RP-HPLC) in combination with a mass spectrometer, a method commonly used for characterization or separation, was employed to identify phenolic, flavonoid, and other compounds. The instruments allow for extract evaluation to be performed in positive and negative modes. The percentages of polyphenol families present in the extracts were determined using molecular weights of over 1200 samples from the Phenol-Explorer database (Material and Methods), as shown in Figure 1. For both rhizome and leaf extracts, the predominant family was found to be phenolic acids, followed by flavonoids.

Table 3 ranks the compounds with the highest relative abundance in the two *MS* extracts. In the case of the RAE extract, the compound with the highest relative abundance was pyrogallol, a furanocoumarin, with a reading of 9.32%. This molecule is found primarily in the rhizomes of various plants [19,20]. It was followed by the flavanone isoxantohumol with a relative abundance of 9.09%. Thus, the family with the highest relative abundance was flavonoids, which were found to be the main polyphenols in this extract.

In the leaf extract (HAE), the compound with the highest relative abundance was p-coumaroyl tartaric acid, a derivative of hydroxycinnamic acid, with a relative abundance of 28%. The relative abundance of daidzein in the HAE extract was 10.85%, with daidzin 7-O-glucosidehe being the only daidzein derivative detected.

### 2.2. Radical Scavenging Capacity of MS Extracts

#### Radical Scavenging Capacity Measurement and Sun Protection Factor (SPF) Evaluation

The radical scavenging capacity of both extracts was evaluated using the DPPH radical molecular inhibition method and the Oxygen Radical Absorbance Capacity (ORAC) method, both commonly used to measure the radical scavenging capacity of substances.

Table 4 shows the results of DPPH and ORAC radical scavenging capacity measurements in *MS* extracts. GAE and vitamin C were used as positive controls in the DPPH inhibition assays and TROLOX in the ORAC assay.

Natural compounds such as phenols and flavonoids have generated recent interest as potential sunscreen ingredients with radical scavenging capacity because of their absorption in the UV-B region. To correlate radical scavenging capacity with UV protection factor, we evaluated the SPF potential of the extracts using a common in vitro spectroscopic method that measures wavelengths between 280 nm and 400 nm [21]. In this study, SPF was obtained for the extracts dissolved in ethanol by running from 292.5 to 320 nm at 5 nm intervals [21]. The obtained measurements of the SPF factor are shown in Table 5 and reported as SPF detected. The equivalent SPF values and SPF levels according to ISO 24443 are also shown. RAE was found to have a moderate SPF level, and HAE, a moderately high SPF level.

### 2.3. Antimicrobial Assays

#### Minimum Inhibitory Concentration

In order to determine the capacity of both extracts to inhibit the growth of *S. aureus* and *S. epidermidis*, the minimum inhibitory concentration test was performed.

The RAE and HAE extracts tested on microorganisms showed no microbial sensitivity to either of the two extracts at the concentrations evaluated in this study, as shown in Table 6.

Although neither extract had inhibitory effects on microbial growth, they were able to interfere with bacterial biofilm formation. The 48-h incubation of microorganisms with RAE and HAE *MS* extracts at concentrations ranging from 0 to 512 μg/mL showed a dose-dependent inhibitory effect of the extracts on biofilm formation. This effect was more pronounced in the RAE extract against *S. epidermidis*; however, both extracts showed inhibitory effects against both bacteria. In biofilm disruption assays, higher concentrations of RAE and HAE were found to have a more pronounced effect against *S. epidermidis* than *S. aureus*, as shown in Figure 2B,C.

### 2.4. Inhibition of COX Enzymes

The COX-1 enzyme is constitutively expressed and is considered to be a maintenance protein responsible for the physiological functioning of the cell. COX-2, on the other hand, is an inducible enzyme, and its expression is activated when tissue damage or inflammatory conditions occur. Inhibition of COX-1 results in side effects such as platelet aggregation inhibition and melanoma formation [22], while inhibition of COX-2 has therapeutic effects on inflammation-induced pain [23]. Because of the importance of these enzymes for the human organism, the inhibitory effects of the extracts on both enzymes were evaluated. Table 7 shows the IC_50_ values for the effect of *MS* extracts on the COX enzymes. The results indicate that both extracts selectively inhibit (selectivity index) the COX-2 and COX-1 enzymes, but RAE has higher selectivity index than HAE. The positive controls used were SC560 for COX-1 and Celecoxib for COX-2 [24].

### 2.5. Cell Line and Treatments

#### 2.5.1. HDFa Cell Line Cytotoxicity Assay

The cytotoxicity of both extracts was evaluated on the HDFa cell line using the MTS assay measuring the formazan formation, allowing for the determination of the mitochondrial functionality of the treated cells. The results showed that the extracts used in this study (1 to 100 μg/mL) had no detrimental effect on dermal cells (Figure 3).

#### 2.5.2. Evaluation of Cytotoxicity to HDFa in Infections Caused by *S. aureus* and *S. epidermidis*

HDFa cells were infected at an MOI of 5; the concentrations used are shown in Figure 4. The release of the LDH enzyme was used as a marker of damage to the cell membrane [25]. Figure 4A shows that when cells were incubated with *S. aureus*, LDH release increased by almost 60% at 6 h post infection, indicating cell death. However, this effect was markedly diminished when cells were incubated simultaneously with *MS* extracts and the bacteria.

Less LDH release was observed in assays in which *S. epidermidis* and *MS* extracts were simultaneously in contact with cells.

#### 2.5.3. Formation of Reactive Oxygen Species in the HDFa Cell Line

Reactive oxygen species (ROS) are a key marker of inflammation pathway and macromolecular damage. To evaluate the effect of the *MS* extracts on cell membrane damage and inflammation, ROS formation was measured generating damage using an MOI of 5 *S. aureus* or *S. epidermidis* using the H2DCFDA-DCF probe. To this end, the HDFa cell line was incubated with *MS* for 3 h (Figure 5A). It was determined that the extracts tended to slightly reduce normal ROS levels. When HDFa cells were simultaneity exposed to *S. aureus* and *MS* extracts for 3 h of incubation, a significant reduction in intracellular ROS formation was observed compared to ROS generated in the presence of *S. aureus* without the extracts (Figure 5B). Cells treated with 63 µg/mL RAE generated 76.88% ROS. However, when HDFa cells were preincubated with the extracts for 3 h and then infected with *S. aureus* or *S. epidermidis* and incubated for an additional 3 h, the extracts also reduced ROS formation. The best result was obtained with 100 µg/mL RAE extract compared to 63 µg/mL RAE extract, with the HAE extract at the same concentrations (Figure 5B).

## 3. Discussion

There is a wealth of information on the medicinal properties of various plants, including radical scavenging capacity, anti-inflammatory, and antimicrobial effects, which are useful in the treatment of many diseases. These properties are associated with phytochemicals such as polyphenols, flavonoids, and triterpenes. However, there are very few reports on the properties of *MS*. In this paper, we studied two *MS* extracts (rhizome and leaf) and found that the RAE and HAE extracts had different contents of secondary metabolites. For example, the RAE extract had a very high content of polyphenols and flavonoids per gram of dry sample. Plants are known to contain high concentrations of polyphenols as a defense mechanism against pathogens [26].

It has been reported in the literature that ethyl acetate extracts are rich in phenolic components [27,28,29,30,31]. We found that more than 50% of the phenolic content of the RAE and HAE extracts consisted of flavonoids. Similar results have been described by other authors [30,32].

The amounts of phenols and flavonoids have been previously reported in *MS* from Tamil Nadu in India [33], but there are no studies on *MS* from Rapa Nui. The main methods used for the quantification and characterization of extracts from plants are the spectrophotometric and chromatographic methods. In this study, the identification of phenols, flavonoids, and other compounds was performed by the RP-HPLC-*MS*/*MS* method, while concentrations were analyzed by the spectrophotometric method. Figure 1 shows that phenolic acids were the predominant class, accounting for over 45% in both extracts. These compounds correspond mainly to the derivatives of hydroxybenzoic and hydroxycinnamic acids, as has been observed in several ferns (Appendix A) [34,35]. The second class is flavonoids and other types of polyphenols, with an emphasis on coumarin derivatives, as previously reported for *MS* extracts from Mahanoro, Madagascar [1]. Similar results have been reported for the edible fern *Diplazium esculentum* from India [36], whose leaves contain about 40% flavonoids. In addition, a variety of hydroxycinnamic acids were found mostly in the leaf samples of the fern *Stenochlaena palustris* from Malaysia [37].

The flavonoids found in this study correspond to both flavonoids and their glycosides, presumably luteolin, kaempferol, isorhamnetin, and their derivatives. The average molecular mass of flavonoids is 345 g/mol, shifting toward 280 g/mol (Appendix A). Flavonoids with a basic structure have molecular weight of 222.24 g/mol, similar to phenolic acids, and the majority of the flavonoids found were glycosylated. Members of the stilbene group (Appendix A), resveratrol (negative mode in the HAE extract and positive mode in the RAE extract), and pterostilbene (positive mode in HAE) were also detected. Stilbenes are produced in a variety of plant species, including *Vitis vinifera*, red berries, and peanuts [38,39]. Studies have demonstrated the important role of resveratrol in human health.

We found that 44% of the phenolic acids detected belonged to the hydroxycinnamic acid subclass and 44% to the hydroxybenzoic acid subclass. The average weight of the phenolic acid found was 277 g/mol, shifting toward 200 g/mol. The molecular weights of phenolic acids ranged from 120 to 220 g/mol, and the phenolic acids found in this study were predominantly glycosylated. Glycosylated molecules tend to be more bioavailable upon ingestion due to their altered bioavailability properties, membrane disintegration, bioactivity, and metabolic stability [40].

The other types of polyphenols were mostly coumarin derivatives, e.g., sculetin and isopimpinellin. This class of polyphenols has a broader range of molecular masses, ranging from low molecular weight molecules such as pyrogallol (126.11 g/mol) to more complex molecules such as 5-heneicosenylresorcinol (402.7 g/mol).

The extract with the best radical scavenging capacity in the DPPH inhibition assay was RAE, with an inhibition rate of 82.96%. The observed values are consistent with the results described by Ding [41], who analyzed 31 fern extracts from Asia and found that the IC_50_ ranged between 5.9 µg/mL and 133.2 µg/mL, with an average of 39.47 ± 35.92 µg/mL. Our results showed that the IC_50_ values of the RAE and HAE extracts in the DPPH inhibition assay were 12.05 µg/mL and 20.34 µg/mL, respectively, which is close to the values reported by Ding [41]. In the ORAC test, the highest radical scavenging capacity was demonstrated by the RAE extract, with an ORAC value of 1.63 (Table 4).

There is evidence to suggest that *MS* extracts have a cytoprotective effect against UV-B ray damage to human epithelial cells [42]. Thus, *MS* may have both an epithelial protective effect at the cellular level and an effect as a sunscreen filter at the dermal level. Indeed, HAE extracts were observed to have an SPF of 20 (medium-high level) and RAE an SPF of 15 (medium level). Therefore, it is quite exciting to consider the possibility that some *MS* compounds may find their way into cosmetic products.

Based on the “Performance standards for antimicrobial susceptibility tests” [43] of the “Clinical and Laboratory Standards Institute” (CLSI), *S. aureus* can be resistant to *MS* extracts and susceptible to kanamycin and chloramphenicol; *S. epidermidis* is resistant to *MS* and kanamycin extracts and moderately susceptible to chloramphenicol (Table 6). Even though the extracts did not show strong antimicrobial activity, they were effective in inhibiting and disrupting biofilms formed by the bacteria *S. aureus* and *S. epidermidis* (Figure 2). Bacteria of the genus *Staphylococcus* are recognized as the most frequent causes of infections involving biofilm formation. Skin infections in humans are frequently colonized by commensal bacteria such as *S. aureus*. These bacteria cause epithelial dysbiosis and increase biofilm formation [44].

Assays with 512 µg/mL pyrogallol show an MIC value against *S. aureus* [43] without compromising the integrity of the MO membrane [45,46], an effect also observed with resveratrol [47]. Chin [48] reported that daidzein and daidzin have lower MIC values than pyrogallol against *S. aureus*, with daidzein being more effective than daidzin. Daidzeinhad an MIC concentration of 64 µg/mL and the daidzin of 128 µg/mL [48]. These results suggest that the total polyphenol concentration in the samples is not relevant for the antimicrobial activity, indicating that the activity may have decreased due to the presence of other molecules in the extracts. The results also show that *S. epidermidis* is more sensitive to *MS* extracts than *S. aureus*, due to the presence of other components in the extracts, such as cirsimaritin [49] and resveratrol [50]. Our own results were similar, as both extracts had an effect on *S. epidermidis* but not on *S. aureus*.

The assays performed to evaluate biofilm inhibition and disruption depended on the MIC concentration of each extract tested on the bacteria. In this study, MIC values and three serial dilutions were used for the assay. If the extract did not exhibit an MIC value, a concentration of 512 µg/mL of the extract was used. Figure 2A,B show graphs of bacterial biofilm inhibition and disruption by the RAE extract against *S. aureus* and *S. epidermidis*. In this case, the RAE extract showed significant inhibition of biofilm formation (close to 50% in the assay with 512 µg/mL) against *S. aureus*. These results may be attributed to the relative abundance of polyphenols such as isoxanthohumol, resveratrol, and kaempferide [51,52]. These types of compounds have been shown to have the ability to inhibit biofilm formation by downregulating the expression of genes such as rsbU and spa, which are genes responsible for bacterial adhesion, communication, and bacterial protection [53,54,55]. The extract that showed more significant inhibition of biofilm formation was RAE at 128 µg/mL against *S. epidermidis*.

Both extracts analyzed showed a dose-dependent effect in biofilm disruption. The results suggest that in assays against *S. aureus*, these extracts may inhibit bacterial biofilm formation by: (i) increasing susceptibility to antibiotics; (ii) creating instability in the bacterium. Therefore, in the future, it might be interesting to evaluate the effect of extracts on different concentrations of antibiotics in biofilm studies and to assess the possibility of reducing the antibiotic concentrations currently used to treat infections caused by these bacteria, as well as to consider the possibility of synergistic effects of the two compounds.

The inhibition assay showed a greater reduction in the percentage of biofilm. Biofilm formation consists of four steps: i.—bacteria attach to a surface; ii.—microcolonies are formed; iii.—biofilm maturation; and finally, iv.—bacteria spread to colonize other surfaces [53]. In the biofilm inhibition test, the bacteria are in steps i and ii, while in the biofilm disintegration test, the bacteria are in step iii. In this step, the barriers that constitute a defense mechanism are already present, requiring a more complex process of elimination. Assays with *S. epidermidis* (Figure 2B,D) confirmed that the rhizome extract inhibited biofilm formation more efficiently than the leaf extract. The HAE extract was effective in disrupting biofilms rather than inhibiting their formation. This extract is known to contain resveratrol, and it has been reported that this compound inhibits bacterial quorum sensing and disrupts bacterial biofilms of the genus *Staphylococcus* [54].

The different responses of the two bacteria to the evaluated extracts can be mainly attributed to multidrug resistance and the presence of virulence genes in both pathogens. *S. epidermidis* exhibits multidrug resistance [55,56,57], whereas *S. aureus* has major resistance mechanisms, making it a highly virulent bacterium tolerant to various antibiotics due to increased activation of these genes [58].

The *MS* extracts had no effect on *S. aureus* viability, but they had an effect on the formation and disruption of bacterial biofilm. It has been reported that phenolic compounds do not play a predominant role in the viability of many bacteria, but they may play a role in reducing virulence, deactivating quorum sensing, and reducing the production of α- family proteins. It has been observed that the main inhibitors that can inactivate the metabolism of bacteria [59], their adhesion to surfaces [52,60] and inhibit biofilm formation, are phenolic compounds [61,62].

The results of the inhibition of COX enzymes were complemented by the DPPH radical scavenging assays and ORAC assays, because both extracts not only have the ability to act on radical molecules, but also to inhibit their production. Although not all extracts exhibited a low IC_50_ for COX-2 enzyme (the results were compared with the Celecoxib drug used as positive control), they also did not show a selectivity index of less than 1. The HAE extract exhibited an IC_50_ of 3.52 µg/mL for COX-2 enzyme and a selectivity index of 6.84. This extract was the second-best in terms of radical scavenging ability. The RAE extract showed a IC_50_ of 3.14 for COX-2 and a selectivity index of 9.96. In addition, it had the highest radical scavenging capacity in the DPPH and ORAC assays (Table 4). These results are consistent with the low percentages of ROS production observed when the HDFa cell line was exposed to both bacteria. Therefore, both extracts had a protective effect on the cell line (Figure 4).

Molecules such as resveratrol, present in *MS* extracts, inhibit not only COX-2 expression, but also its activity [63]. Diadzein, which selectively inhibits COX-1 and COX-2 enzyme expression [64], is a compound mainly present in HAE extracts and provides excellent selectivity (Table 7). Flavonoids such as kaempferol, luteolin, and apigenin have the ability to selectively inhibit COX-1 and COX-2 [36,65]. Extracts with a higher concentration of flavonoids were found to exhibit greater selectivity against COX enzymes (Table 7).

None of the extracts evaluated had any detrimental effect on cell viability, as shown in Figure 3. Studies performed with *MS* ethanolic extracts from Tahiti showed that they do not induce cytotoxicity on epithelial cell lines in the range of 1 to 250 µg/mL [42]. Phenolic compounds of the flavonoid family, such as cirsimaritin [66] and kaempferol [67] have been reported to increase cell viability. These compounds are present in the *MS* extracts studied.

The results of the infection assays indicate the possibility that the molecules present in the extracts affect the proliferation of *S. epidermidis* and create a molecular cascade that helps to fight the infection, either by reducing biofilm formation, biofilm adherence to the cell, and the production of virulent proteins from this pathogen [68], or by increasing the viability of HDFa cells, as shown in Figure 3. A large difference was observed in the LDH release among the cells infected with the bacteria, but without the presence of the extracts (Figure 4A,B). These results confirmed the protective effect of the extracts on cells damaged by pathogens.

Previous reports showed that daidzein inhibits the expression of caspases 3 and 9 in viral infections in vitro [69], while pyrogallol inhibits bacterial infections in vivo [70]. In addition, a decreased release of LDH and a decreased expression of IL-1β were observed in bacterial infections, suggesting that these compounds prevent an increase in the oxidative stress produced by infections [71].

The differences between *S. aureus* and *S. epidermidis* in the rate of LDH release may be due to the fact that *S. aureus* interacts more aggressively with the host cell and releases exoenzymes into the intracellular medium (SpIF proteases were detected), thus disrupting the metabolism of the eukaryotic cell [58]. Meanwhile, *S. epidermidis* is characterized by accompanying *S. aureus* as an opportunistic pathogen, generating a bacterial biofilm more rapidly, and being a reservoir of multidrug-resistance genes [72,73].

The reduction in the percentage of ROS may be a product of infection inhibition or the extracts may have exerted an intracellular effect, reduced the production of ROS or decreasing their concentration. *S. aureus* produces Dps proteins, which are responsible for resistance to the nitric oxide and hydrogen peroxide produced by the infected cell [74,75], and are a mechanism of DNA protection. The subsequent reduction in ROS forces the infected cells to produce higher concentrations of ROS to defend themselves against pathogens.

As shown in Figure 5E, preincubation of cells with *MS* extracts produced a protective effect on the cells, possibly helping to avoid infection by *S. epidermidis*. The RAE extract (87.35% less ROS in a concentration of 100 µg/mL) had the lowest rate of ROS formation in the presence of bacteria. These results suggest either a higher expression of antioxidant proteins, such as the enzymes glutathione peroxidase and superoxide dismutase [76], or a suppression of pro-oxidative proteins, such as COX-2.

The ability of a wide variety of flavonoids and stilbenes to inhibit pro-oxidative pathways and suppress the COX-2 enzyme has been previously reported [77]. *MS* extracts have also been reported to protect against oxidative stress by activating molecular cascades involved in signal transduction, stress, and extracellular matrix synthesis and repair [42]. It has been reported in our laboratory and in the literature that polyphenols in *MS* extracts reduce the concentration of ROS produced by oxidative stress, as is the case with resveratrol [78] and pyrogallol [79]. Daidzein and diadzin can reduce ROS concentrations and LDH release in response to oxidative damage, raising the activity of the enzyme superoxide dismutase and lowering the expression of COX-2 and NF-κβ [64,80,81,82]. Thus, when HDFa cells were pre-incubating with 100 µg/mL RAE and then infected with *S. aureus*, a 64% reduction in ROS formation was observed relative to infected controls (Figure 5B,C).

Coumarin derivatives such as protocatechuic acid, psoralen, and catechol in *MS* extracts have been reported to inhibit NF-κβ, reduce ROS, and improve cell survival [12,71].

## 4. Materials and Methods

### 4.1. Plant Materials, Cells, and Reagents

Plant species were brought from the CONAF conservation reserve on Rapa Nui. Upon receipt in the Laboratory of Biomedicine and Biocatalysis of the Pontifica Universidad Católica de Valparaíso, rhizomes and leaves were weighed, washed in sterile water, chopped, and stored at −80 °C. Adult Human Dermal Fibroblasts (HDFa) were obtained from Thermo Fisher Scientific (Thermo, Waltham, MA, USA). Bacterial strains *S. aureus* (ATCC 25955) and *S. epidermidis* (ATCC 35984) were purchased from Microbiologics^®^ (Microbiologics, MN, USA). The LDH Cytotoxicity Assay Kit was purchased from Thermo Scientific (Thermo, Waltham, MA, USA). Gallic acid, TROLOX, AAPH, n-Hexane, ethyl acetate and DPPH were purchased form Sigma-Aldrich (St. Louis, MO, USA). Culture medium 106, LSGS supplement, antibiotics penicillin and streptomycin, Trypticase Soy Agar (TSA), and Trypticase Soy Broth (TSB) were purchased from Oxoid, Thermo Fisher Scientific (Thermo, Waltham, MA, USA). All other reagents used in this study were purchased from Merck Co. (Kenilworth, NJ, USA).

### 4.2. Cell Growth Conditions

The HDFa cells were cultured in 106 medium supplemented with LSGS, penicillin (50 U/mL), and streptomycin (50 mg/mL). Cells were stored at 37 °C in a humid atmosphere with 5% CO_2_ for use in subsequent experiments. Bacterial strains S. *aureus* and *S. epidermidis* were routinely grown in TSA and TSB media at 37 °C for 12 to 24 h. The strains were preserved at −80 °C in 15% glycerol.

### 4.3. Determination of MS Phytoextracts Metabolite Composition and Radical Scavenging Capacity

#### 4.3.1. Extract Preparation

A sample of 20 g leaves or rhizomes was washed with distilled water. The leaves and rhizomes were then dried, chopped, and frozen at −80 °C. After freezing, both leaves and rhizomes were triturated separately, and the plant parts were ground and treated with hexane (500 mL) under magnetic stirring for 72 h at 35 °C. The extracts were then filtered through filter paper (Whatman No. 1); hexane was removed by incubation at 40 °C. Next, both extracts were rotary evaporated (Heildolph, Schwabach, Germany) at 40 °C under reduced pressure, ethyl acetate was added, and the mixture was stirred at 35 °C for 72 h. Finally, the solution was filtered and the solvents were removed with a rotary evaporator (Heildolph, Schwabach, Germany). The dried samples were weighed, dissolved in ethanol at a concentration of 1000 µg/mL, and stored at −20 °C until use. Polyphenol extracts from rhizomes and leaves were prepared using ethyl acetate. The designation of the rhizome extract was abbreviated as RAE and that of the leaf extract as HAE.

#### 4.3.2. Determination of Total Phenolic Compounds

Folin–Ciocalteu reagent was used to quantify the total polyphenol content in the leaves and rhizomes. For this purpose, ethanolic extracts were prepared from leaves and rhizomes as described by [83]. Briefly, 100 µL of the leaf and rhizome extracts (RAE, RH, HAE and HH) were diluted 1:10 with water and 125 µL of Folin–Ciocalteu 1 N reagent was added and shaken vigorously. Then, 625 µL of 20% Na_2_CO_3_ was added and the samples were shaken again for two h. Finally, absorbance was measured at 760 nm on an HPUV 8453 spectrophotometer (Agilent, Santa Clara, CA, USA). Absorbance values were interpolated using a gallic acid standard curve (0–10 mg/L), and total phenolic content was expressed as mg gallic acid equivalents (GAE) per gram of dried extract. Experiments were performed in triplicate.

#### 4.3.3. Determination of Total Flavonoid Compounds

Determination of total flavonoids was performed using the methodology of Liu [84] with modifications. Briefly, 30 µL sodium nitrite (10% *w*/*v*), 60 µL aluminum chloride hexahydrate (20% *w*/*v*), 200 µL NaOH (1 M), and 400 µL distilled water were added to 100 µL of the sample. Absorbance was recorded after 5 min at 415 nm. The results were interpolated on a Kaempferol calibration curve. The results were expressed in mg of kaempferol (KE) per gram of dry extract. The experiment was conducted in triplicate.

#### 4.3.4. Radical Scavenging Capacity Determined by the DPPH Assay

The radical scavenging activity of the extracts was evaluated by the DPPH (1,1-diphenyl-2-picrylhydrazyl) assay [85]. Briefly, 1 mL of 0.1 mM DPPH radical solution in ethanol was mixed with 50 µL of leaf, rhizome extracts, gallic acid or vitamin C in a concentrations of 20 µg/mL. DPPH is reduced by antioxidants, causing a color change from purple to yellow. The color change was measured by absorbance (Abs) at 518 nm after 20 min of reaction using an Epoch ELISA reader (ELx800, BioTek, Winooski, VT, USA). The DPPH inhibition percentage was calculated using the following equation:(1)$$\%\mathrm{radical}\;\mathrm{scavenging}\;\mathrm{activity}=\left(\frac{\mathrm{abs}\;\mathrm{control}-\mathrm{abs}\;\mathrm{sample}}{\mathrm{abs}\;\mathrm{control}\;}\right)\times 100$$
where, Abs control is the absorbance of DPPH in the absence of a sample and Abs sample is the absorbance of DPPH in the presence of a sample or standard. The radical scavenging capacity of the extracts using DPPH was expressed as mg GAE equivalents per gram of sample dry weight (expressed as the concentration of the sample required to reduce the DPPH absorbance by 50% = IC_50_). The IC_50_ values were calculated by linear regression of the plots. These experiments were performed in triplicate.

#### 4.3.5. Radical Scavenging Capacity by the ORAC-FL Assay

The ORAC value was measured according to the method described in [86] with modifications [87]. The reaction was performed in sodium phosphate buffer (75 mM, pH 7.4) using black-walled 96-well plates in a final volume of 200 µL. Twenty µL of each extract or TROLOX (20 µg/mL) and fluorescein solutions (120 µL; 70 nM, final concentration) were placed in each well of the microplate. The mixture was preincubated for 15 min at 37 °C. The AAPH solution (60 µL; 12 mM final concentration) was added rapidly and the microplate was immediately read using a fluorescence reader (Synergy HT multi-detection microplate reader; Bio-Tek Instruments, Inc., Winooski, VT, USA). Fluorescence was recorded every minute for 80 min from normalized curves and the area under the fluorescence decay curve (AUC) was calculated as follows:(2)$$\mathrm{AUC}=1+\overset{i=80}{\underset{i=1}{\sum }}\frac{f_{i}}{f_{0}}$$
where, f_0_ is the initial fluorescence reading at 0 min and f_i_ is the fluorescence reading at time i. The AUC of a sample was calculated by subtracting the AUC of the blank space. Regression equations between net AUC and antioxidant concentration were calculated for all the samples. The ORAC-FL values were expressed as Trolox equivalents using the standard curve calculated for each assay. The experiments were performed in triplicate.

#### 4.3.6. The RP-HPLC-*MS*/*MS* Analysis of the *MS* Extracts

The RP-HPLC-*MS*/*MS* analysis was performed using the HPLC 1100 equipment (Agilent, Santa Clara, CA, USA) and a TRAP 3200 Q TRAP hybrid triple quadrupole/linear ion mass spectrophotometer. A gradient of solvents A (0.1% formic acid) and B (100% methanol) was used, with a flow rate of 0.5 mL/min. Solution B was graduated from 5 to 50% over 30 min, followed by an increase to 75% of B over 25 min. The measurement range was between m/z 100 and 1000 and was detected in positive and negative mode. Data were analyzed using the Thermo Xcalibur SP1.48 program (version 2.2, Thermo Fischer Scientific, Waltham, MA, USA), using molecular weights from http://phenol-explorer.eu/ (version 3.6) (accessed on 20 June 2021).

#### 4.3.7. Determination of Sun Factor Protection (SPF) by Ultraviolet Spectroscopy of the *MS* Extracts

To evaluate SPF, samples were diluted with ethanol to 1 mg/mL and filtered. The absorbance spectra of the samples in solution were obtained in the range of 290 to 320 nm, every 5 nm, with a spectrophotometer HPUV 8453 (Agilent, Santa Clara, CA, USA), using a 1 cm quartz cuvette and ethanol as a blank, and each determination was made in triplicate. The SPF determination was calculated using Mansur’s equation [88]:(3)$$\mathrm{SPF}=\mathrm{CF}\;\times \overset{320}{\underset{290}{\sum }}\mathrm{EE}\left(\lambda \right)\times \;I\;\left(\lambda \right)\times \;\mathrm{Abs}\;\left(\lambda \right)$$
where, EE: Erythemal effect spectrum. I: Solar intensity spectrum. Abs: Absorbance of the sample. CF: Correction factor.

### 4.4. Antimicrobial Assays

#### 4.4.1. Determination of Minimum Inhibitory Concentrations (MIC)

MIC_80_ was evaluated for both microorganisms in the presence and absence of the *MS* extracts. To this end, a McFarland value of 0.5 was seeded on Müller Hinton Broth medium in a 96-well curved bottom plate, with a range of 0.060 µg/mL to 512 µg/mL of the *MS* extracts. The microorganisms were allowed to grow under agitation at 150 rpm and a temperature of 37 °C. Growth inhibition was recorded after 24 h at 600 nm on an EPOCH spectrophotometer.

#### 4.4.2. Biofilm Formation Inhibition Assay

The biofilm inhibition assay was performed as described in [89] with some modifications. A McFarland value of 0.5 was seeded in Müller Hinton Broth medium in a 96-well plate. Bacteria were treated with the MIC_80_ concentration corresponding to the extract and three serial dilutions from the MIC. The microorganisms were left under agitation at 150 rpm and at 37 °C for 48 h. Subsequently, the medium was extracted and washed twice with 150 µL PBS, and the plate was dried at 60 °C for one h, stained with 175 µL 0.4% crystal violet in ethanol (molecular biology grade) for 15 min, washed with 180 µL PBS, and dried at 60 °C for 20 min. Then, 200 µL 0.03% acetic acid was added and absorbance was measured at 570 nm on an EPOCH device. The biofilm percentage was calculated from the untreated wells. The test was performed in triplicate at three different times.

#### 4.4.3. Biofilm Disintegration Test

The experiment was performed as described in [89] with some modifications. A McFarland value of 0.5 was seeded in Müller Hinton Broth medium in a 96-well plate. The microorganisms were left under agitation at a stirring speed of 150 rpm and 37 °C for 24 h. Subsequently, they were treated with the concentration of MIC corresponding to the extract and three serial dilutions from MIC_80_, and allowed to grow for a further 48 h. The medium was removed, the bacteria were washed with 150 µL PBS twice, and the plate was allowed to dry at 60 °C for one. Then it was stained with 175 µL 0.4% crystal violet in ethanol for 15 min, washed with 180 µL PBS, and allowed to dry at 60 °C for 20 min. Subsequently, 200 µL 0.03% acetic acid was added and absorbance was measured at 570 nm in an EPOCH device. The percentage of biofilm formation was calculated from the untreated wells. The test was performed in triplicate at three different times.

### 4.5. Inhibition of COX Enzymes

To evaluate the inhibition of the COX enzymes, the BioVisión^®^ “COX-1 Inhibitor Screening Kit (Fluorometric)” and “COX-2 Inhibitor Screening Kit (Fluorometric)” were used according to the manufacturer’s instructions. The inhibition of prostaglandin G2 formation (a product formed from arachidonic acid) by the action of COX enzymes was evaluated. Measurements were made over time by incubating with 3 µg/mL *MS* extracts and the drug SC560 (commercial inhibitor) and measuring the fluorescence (λexc/λem: 535/587 nm) in a multiplate reader Skanit^®^ Appliskan (Thermo Fischer Scientific, Waltham, MA, USA) at 25 °C for 10 min. The percentage of inhibition was calculated using the following formula:(4)$$\%\;\mathrm{Inhibition}=\frac{\mathrm{slope}\;\mathrm{Enzyme}\;\mathrm{control}-\mathrm{slope}\;\mathrm{inhibitor}\;\mathrm{compunds}}{\mathrm{slope}\;\mathrm{enzyme}\;\mathrm{control}}\times 100$$

### 4.6. Cell Line and Treatments

#### 4.6.1. HDFa Cell Line Cytotoxicity Assay

The cytotoxic activity was evaluated [90] using the immortalized cell line HDFa. For this purpose, 3 × 10^3^ cells were seeded in 96-well plates and incubated with the *MS* extracts for 24 h in 106 culture medium at 37 °C and 5% CO_2_. After incubation, cell viability was determined using the MTS Cell Viability Assay. It was measured by spectrophotometry at 490 nm.

#### 4.6.2. Cytotoxicity Evaluation of the *MS* Extracts on the HDFa Cell Line in *Staphylococcus aureus* and *Staphylococcus epidermidis* Infection

The cytotoxic effect of the *MS* extracts on the HDFa cell line infected by *S. aureus* and *S. epidermidis* was evaluated (independently). For this purpose, 271,500 cells/well were seeded in a 48-well plate and incubated for 24 h in 106 culture media, at 37 °C with 5% CO_2_. Bacteria were added with an MOI of 5, i.e., each seeded cell was infected with 5 bacteria. Cell viability was evaluated after the release of lactate dehydrogenase (LDH) using the LDH Cytotoxicity Detection Kit (Takara Bio USA, Inc., San Jose, CA, USA, a specialized detection kit for eukaryotic cells. Aliquots of the medium were taken at 3 and 6 h of incubation with the microorganisms and the following formula was used to determine the percentage of cell viability:(5)$$\%\;\mathrm{Citotoxicity}=\frac{\mathrm{treated}\;\mathrm{cells}\;-\mathrm{untreated}\;\mathrm{celles}}{\mathrm{damage}\;\mathrm{contrl}-\mathrm{untreated}\;\mathrm{cells}\;}\times 100$$

#### 4.6.3. Evaluation of the Formation of Reactive Oxygen Species

To determine the percentage of ROS formation, 5 × 10^3^ cells/well were preincubated with culture medium for 24 h. Different concentrations of the *MS* extracts to be analyzed were added to each well and incubated for 3 h. Then, infection with *S. aureus* and *S. epidermidis* was performed (independently) for 3 h. In addition, the compounds and bacteria were incubated on the HDFa cells for 3 h. After this time, the medium was changed to KHB and the H2DCFDA-DCF probe was added at a concentration of 25 µM and incubated for 30 min at 37 °C. The formation of ROS was measured by evaluating the fluorescence intensity of the oxidation product of the probe, which fluoresces at λexc/λem: 490/525. These kinetics were evaluated in a multiplate reader Skanit^®^ Appliskan (Thermo Fischer Scientific, Waltham, MA, USA) at 37 °C for half an h while protected from light.

### 4.7. Statistical Analysis

Data were expressed as mean ± standard deviation. Differences between experimental groups were analyzed using Student’s *t* and ANOVA (followed by Tukey’s posttest), according to the experimental protocol. Differences were considered significant at *p* < 0.001. For data analysis, the statistical software SPPS (Version 17; SPSS Inc., Chicago, IL, USA) and Graphpad Prism (version 7.04 GraphPad Software, La Jolla, CA, USA) were used.

## 5. Conclusions

*Microsorum scolopendria* exhibited high concentrations of polyphenols, especially in the rhizomes, consisting primarily of flavonoids. A wide variety of polyphenols was found in the *MS* extracts, with protocatechic acid 4-O-glucoside, cirsimaritin, isoxanthohumol, daidzein, pyrogallol, and resveratrol exhibiting the highest relative abundances. The polyphenols may be interesting from a pharmacological perspective because of their high radical scavenging activity and ability to modulate intracellular metabolic pathways.

The RAE extract exhibited higher radical scavenging activity in the DPPH and ORAC assays. The RAE and HAE extracts showed impressive SPF levels, with the HAE extract proving to be the best with a moderately high SPF level due its resveratrol, ferulic acid, and catechol content, among other compounds with structures that can absorb UV rays.

The evaluation of COX enzyme inhibition showed that both *MS* extracts were selective. The RAE extract showed a higher selectivity index, with 9.96 higher inhibition against COX-2 than COX-1.

Although the extracts showed no effect on bacterial viability according to MIC, they did show an effect on the inhibition of biofilm formation for *S. epidermidis* and *S. aureus*. In this respect, RAE showed the best results, as it was able to suppress biofilm formation by over 50%. Finally, the extracts reduced the damage produced by *S. aureus* and *S epidermidis* in an infection assay on the HDFa cell line analyzed for the LDH release. The best protective effect was found with the RAE extract, which was evaluated by measuring the production of ROS induced by microorganisms. This result demonstrated that *MS* has multiple biological properties and therapeutic potential. We suggest that in the future, purified *MS* extract should be analyzed on a model with damaged cells, and the mechanism of action on ROS production and COX-2 regulation should be evaluated in vitro and in silico to find cellular targets and therapeutic agents.

## Figures and Tables

**Figure 1 molecules-27-05467-f001:**
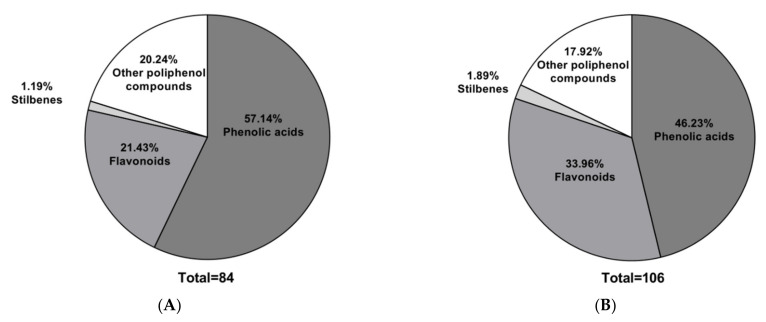
Relative distribution n of polyphenol families in *MS* extracts from (**A**) RAE, (**B**) HAE (unit: %).

**Figure 2 molecules-27-05467-f002:**
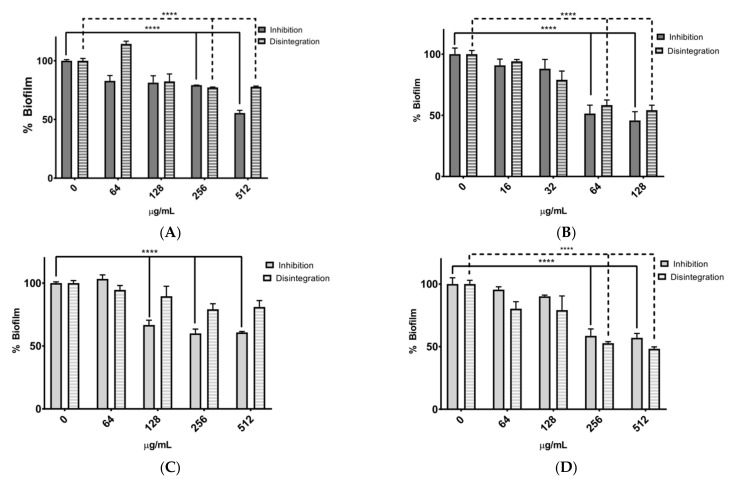
Biofilm formation inhibition and disruption assay for *S. aureus* and *S. epidermidis*. (**A**) RAE treatment against *S. aureus*. (**B**) RAE treatment against *S. epidermidis* (**C**) HAE treatment against *S. aureus*. (**D**) HAE treatment against *S. epidermidis*. **** indicates *p* ≤ 0.0001 between no treatment control and the treated bacteria.

**Figure 3 molecules-27-05467-f003:**
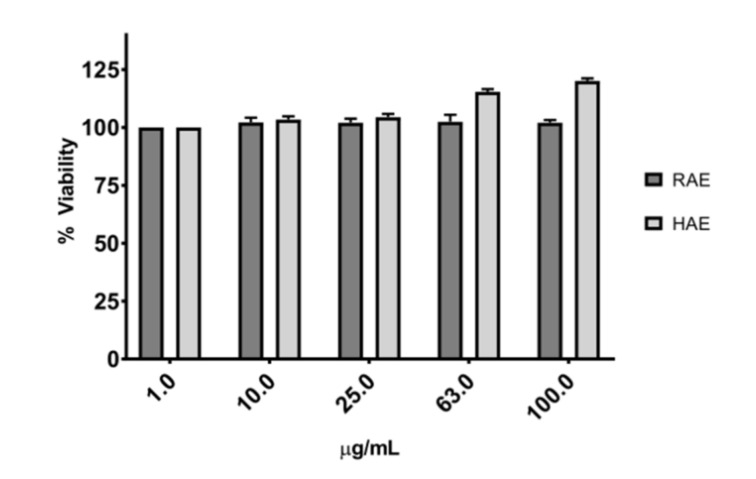
Viability of HDFa cells exposed to *MS* extracts at concentrations of 1 µg/mL to 100 µg/mL for 24 h. These results prompted us to use the RAE and HAE extracts at concentrations between 63 and 100 µg/mL for infection assays on HDFa cells.

**Figure 4 molecules-27-05467-f004:**
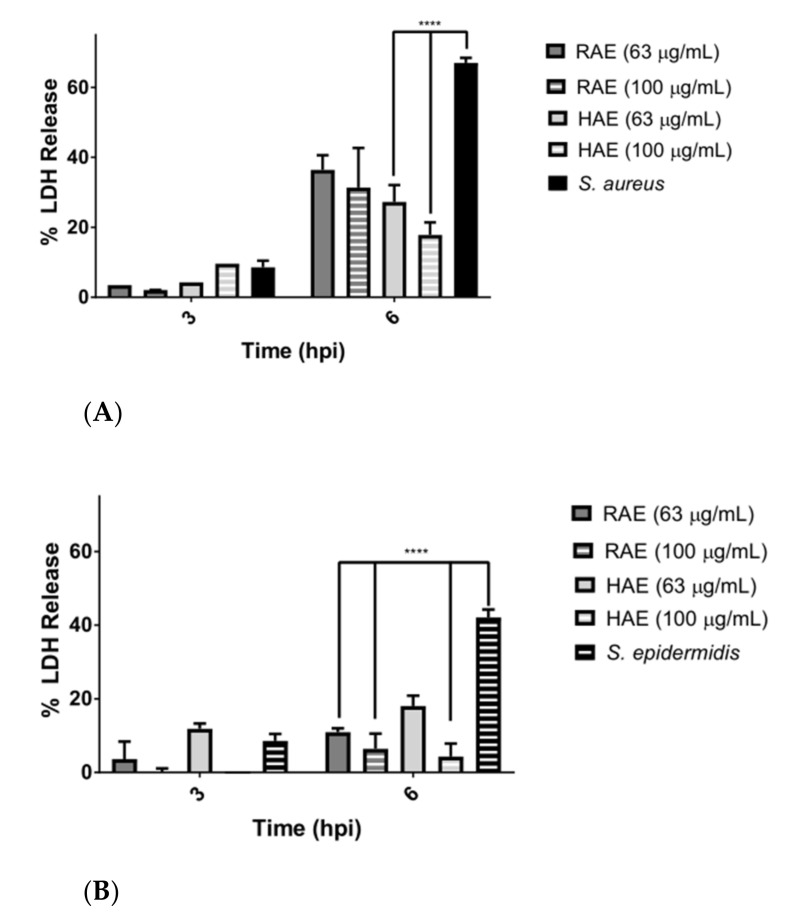
Lactate Dehydrogenase release. (**A**) HDFa cells exposed to an MOI of 5 S. aureus and different concentrations of the RAE and HAE extracts at 6 h post infection. (**B**) HDFa cells exposed to an MOI of 5 S. epidermidis and different concentrations of the RAE and HAE extracts at 6 h post infection. **** indicates *p* ≤ 0.0001 between the respective control of untreated cells and the treated cells.

**Figure 5 molecules-27-05467-f005:**
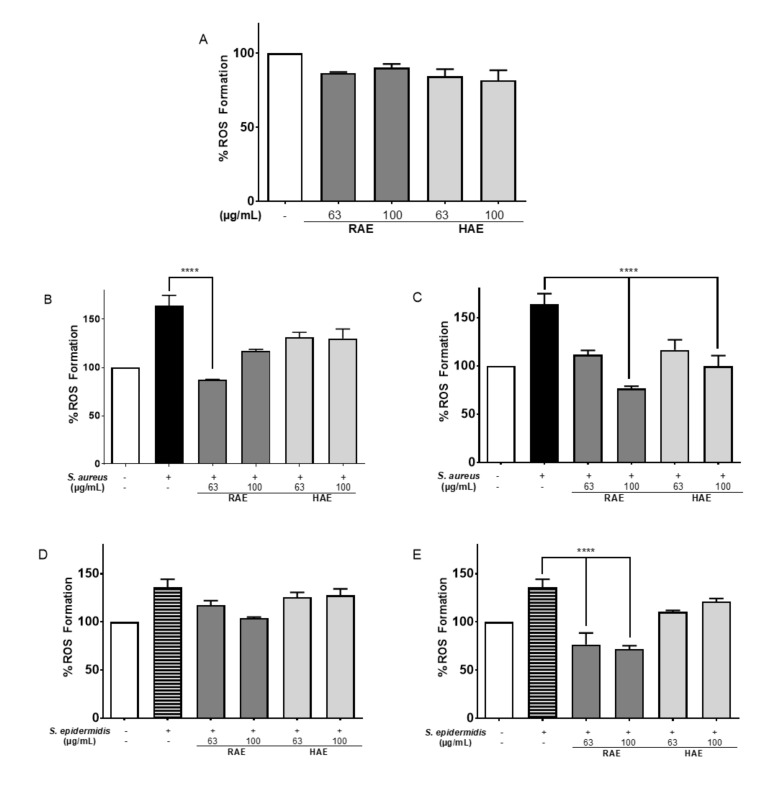
Percentage of reactive oxygen species. (**A**) HDFa cells exposed to *MS* extracts for 3 h. (**B**) Simultaneous exposure for 3 h to S. aureus and different concentrations of *MS* extracts. (**C**) Pretreatment for 3 h with different concentrations of *MS* extracts, and subsequent incubation for 3 h with S. aureus. (**D**) HDFa cells exposed for 3 h to S. epidermidis and different concentrations of *MS* extracts. (**E**) HDF cells pretreated for 3 h with different concentrations of *MS* extracts and subsequently incubated for 3 h with S. epidermidis. **** indicates *p* ≤ 0.0001 between the respective control of untreated cells and the treated cells.

**Table 1 molecules-27-05467-t001:** Weight of extracts from *MS* samples.

Scheme	% Yield	Weight of Extract Obtained (mg/g)
RAE	1.19	1.53
HAE	20.77	2.71

**Table 2 molecules-27-05467-t002:** Total polyphenols and flavonoids in *MS* extracts.

Sample	Total Polyphenols (mg GAE/g Dry Sample)	Total Flavonoids (mg KE/g Dry Sample)
RAE	57.13 ± 0.81	29.01 ± 0.65
HAE	14.77 ± 0.52	10.58 ± 0.56

**Table 3 molecules-27-05467-t003:** Relative abundances of compounds in *MS* extracts.

			Relative Abundance (%)	
Compound	Family	Mode	RAE	HAE
Protocatechuic acid 4-O-glucoside	Phenolic acid	+	1.78	2.19
p-Coumaroyl tartaric acid		−	NA	28
Feruloyl tartaric acid		−	NA	12.8
Kaempferol 3-O-glucuronide	Flavonoid	+	NA	1.41
Cirsimaritin		−	8.45	NA
Kaempferide		−	0.47	NA
Isoxanthohumol		−	9.09	NA
Daidzein		−	NA	10.85
Daidzin		−	0.37	2.19
Resveratrol	Stilbene	+	0.13	1.23
Pyrogallol	Others	+	9.32	1.81
Catechol		+	3.16	2.82
1,4-Naphthoquinone		+	0.34	NA

NA: Not among the most abundant in the extract.

**Table 4 molecules-27-05467-t004:** Radical scavenging capacity of *MS* extracts by DPPH and ORAC assays.

Sample	DPPH (% Inhibition)	ORAC
RAE	82.96 ± 0.53	1.63 ± 0.02
HAE	49.16 ± 1.18	1.54 ± 0.14
Gallic acid	79.24 ± 0.26	1.03 ± 0.14
Vitamin C	70.45 ± 1.34	0.52 ± 0.04
TROLOX	-	1

A concentration of 20 µg/mL of RAE, HAE, Gallic acid, Vitamin C and TROLOX was used to perform the DPPH and ORAC assays.

**Table 5 molecules-27-05467-t005:** SPF measured in *MS* extracts.

Sample	SPF Detected	SPF Equivalent	SPF Level
RAE	17.99 ± 0.02	15	Moderate
HAE	21.91 ± 0.38	20	Moderately high

**Table 6 molecules-27-05467-t006:** Minimum 80% inhibitory concentrations of *MS* extracts in *S. aureus* and *S. epidermidis*.

	*S. aureus*		*S. epidermidis*	
Sample	MIC80 (µg/mL)	Category	MIC80 (µg/mL)	Category
RAE	512<	Resistant	128	Resistant
HAE	512<	Resistant	512	Resistant
Kanamycin	2	Susceptible	512<	Resistant
Chloramphenicol	4	Susceptible	16	Intermediate susceptibility

**Table 7 molecules-27-05467-t007:** IC50 and selectivity indices of *MS* extracts on COX enzymes.

	IC50 (µg/mL)		
Sample	COX-1	COX-2	Selectivity Index
RAE	31.28 ± 0.39	3.14 ± 0.02	9.96
HAE	24.08 ± 1.41	3.52 ± 0.01	6.84
SC560	6.54 × 10^−3^ ± 9.02 × 10^−5^	-	-
Celecoxib	-	1.81 ± 0.02	-

## Data Availability

Not applicable.

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
