# Peer review of "Chemistry and Bioactivity of *Microsorum scolopendria* (Polypodiaceae): Antioxidant Effects on an Epithelial Damage Model"

_molecules, 2022, doi:10.3390/molecules27175467_

Round 1
Reviewer 1 Report
Manuscript by Balad et al. reports an extensive investigation on the polyphenols/flavonoid content of two extracts of Microsorum scolopendria (MS), as well as on a variety of potential properties and bioactivities of the two extracts, namely (antioxidant, which should be renamed radical scavenging, see below), antimicrobial, anti-inflammatory an cytotoxic. Owing to the modest literature available on this plant’s constituents and bioactivity, the study is interesting. However, some issue should be addressed before publication.
1) The DPPH assay is not measuring the antioxidant capacity but it is a titration of the antioxidant content of the extract in the way it is implemented in this study. Therefore, its outcome should be interpreted as radical scavenging capacity or “potential antioxidant capacity”.
2) On expressing both DPPH and ORAC assays’ results it would be important that authors indicate the concentration of extract used in the assay and the concentration of the pure reference compounds (Gallic acid, Vitamin C and Trolox) in footnotes under table 4, otherways the table is not self-explanatory ad interpretation of the results require jumping to the Material and Methods section.
3) The above information (concentration of the reference compounds) is apparently not available in Material and Methods sections 4.3.4 and 4.3.5. Please provide such information clearly otherways the data reported in Table 4 are unusable.
4) In section 2.5.3 the ROS formation assay is presented and discussed without giving any information on how ROS formation was assessed, which renders the whole paragraph void and meaningless. Please indicate the assessment method in the section without forcing the reader to jump to the Material and Methods section to understand the paragraph
5) The same criticism outlined above, apparently persists in most other sections. Although authors have described the experimental protocol in the MM section they should also mention it (at least indicate the principle) in the specific results section, to enable the reader appreciate the results without having to jump continuously to the MM section. Please correct
6) English language would require some minor revision
Author Response
RESPONSES TO COMMENTS FROM REVIEWER 1
We sincerely appreciate the valuable comments of Reviewer 1 (in Black color) and respond (in blue color), accordingly to the questions below:
Point 1: The DPPH assay is not measuring the antioxidant capacity but it is a titration of the antioxidant content of the extract in the way it is implemented in this study. Therefore, its outcome should be interpreted as radical scavenging capacity or “potential antioxidant capacity”.
Response 1: This comment was considered in the new manuscript (lane 21-23-62, 68, 130, 131, 133, 135, 137, 140, 145, 156, 241, 295, 370, 375, 377, 500, 501, 513, 519)
Point 2: On expressing both DPPH and ORAC assays’ results it would be important that authors indicate the concentration of extract used in the assay and the concentration of the pure reference compounds (Gallic acid, Vitamin C and Trolox) in footnotes under table 4, other ways the table is not self-explanatory ad interpretation of the results require jumping to the Material and Methods section.
Response 2: This comment was incorporated in footnotes under the Table 4.
Point 3: The above information (concentration of the reference compounds) is apparently not available in Material and Methods sections 4.3.4 and 4.3.5. Please provide such information clearly otherways the data reported in Table 4 are unusable.
Response 3: We have included the corresponding mention in the text and Materials and Methods. Sections 4.3.4 and 4.3.5
Point 4: In section 2.5.3 the ROS formation assay is presented and discussed without giving any information on how ROS formation was assessed, which renders the whole paragraph void and meaningless. Please indicate the assessment method in the section without forcing the reader to jump to the Material and Methods section to understand the paragraph
Response 4: This comment was incorporated according the reviewer´s suggestion.
Point 5: The same criticism outlined above, apparently persists in most other sections. Although authors have described the experimental protocol in the MM section they should also mention it (at least indicate the principle) in the specific results section, to enable the reader appreciate the results without having to jump continuously to the MM section. Please correct.
Response 5: OK, we have incorporated the principles technique in the specific results section.
Point 6: English language would require some minor revision
Response 6. All english language in the manuscript was reviewer.
.

Reviewer 2 Report
The manuscript entitled «Chemistry and Bioactivity of Microsorum scolopendria (Polypodiaceae): antioxidant effects on an epithelial damage model» is an orginal study aiming to perform a characterization of MS rhizome and leaf extracts (RAE and HAE) about their polyphenolic and flavonoid identity, antioxidant, antimicrobial, and anti-inflammatory properties. For that purpose, the chemical compoistion was analyzed by liquid chromatography (RP-HPLC) method coupled to mass spectrometry. The antioxidant and anti-inflammatory activity were evaluated for DPPH, ORAC, ROS formation and COX inhibition activity assay. The antimicrobial properties were evaluated using an infection model on Human Dermal Fibroblast adult (HDFa) cell line incubated with Staphylococcus aureus and Staphylococcus epidermidis. The subject was well-conducted and experiment design was smartly conceived and perfectly performed and so yielding convincing conclusions. But revision is required to improve the manuscript. below are a few comments and concerns :
- The taxonomic name of the studied plant «Microsorum scolopendria » should be changed to « Microsorum parksii » following a botanical publication (JY Meyer « A note on the taxonomy, ecology, distribution and conservation status of the ferns (Pteridophytes) of Rapa Nui (Easter Island) », Rapa Nui Journal, Vol. 27 (1) May 2013) which stated a taxonomic confusion.
- Secondary metabolites having very interesting biological activity from Microsorum ferns such as ecdysteroids should be mentioned at least in the introduction part as the authors reported at least two related papers dealing on Pacific ferns (Microsorum scolopendria and Microsorum grossum) Tahitian ones within their properties and uses but missed to include these very interting components.
- The word « citotoxicity » (in Material and Method part) should be corrected by « cytotoxicity ».
So, I recommend the acceptation of this paper to be published in Molecules journal after these required revisions.

Author Response
RESPONSES TO COMMENTS FROM REVIEWER 2
We sincerely appreciate the valuable comments of Reviewer 2 (in Black color) and respond (in blue color), accordingly to the questions below:
Point 1: The taxonomic name of the studied plant «Microsorum scolopendria » should be changed to « Microsorum parksii » following a botanical publication (JY Meyer « A note on the taxonomy, ecology, distribution and conservation status of the ferns (Pteridophytes) of Rapa Nui (Easter Island) », Rapa Nui Journal, Vol. 27 (1) May 2013) which stated a taxonomic confusion.
Response 1: This comment was considered in the new manuscript.
Point 2: Secondary metabolites having very interesting biological activity from Microsorum ferns such as ecdysteroids should be mentioned at least in the introduction part as the authors reported at least two related papers dealing on Pacific ferns (Microsorum scolopendria and Microsorum grossum) Tahitian ones within their properties and uses but missed to include these very interting components.
Response 2: This comment was incorporated in the introduction our new version of manuscripts.
Point 3: The word « citotoxicity » (in Material and Method part) should be corrected by « cytotoxicity ».
Response 3: This word was not founded in all the text.
